# Kick Bad Guys Out! Conditionally Activated Anomaly Detection in Federated Learning with Zero-Knowledge Proof Verification

## Abstract

Federated Learning (FL) systems are susceptible to adversarial attacks, where malicious clients submit poisoned models to disrupt the convergence or plant backdoors that cause the global model to misclassify some samples. Current defense methods are often impractical for real-world FL systems, as they either rely on unrealistic prior knowledge or cause accuracy loss even in the absence of attacks. Furthermore, these methods lack a protocol for verifying execution, leaving participants uncertain about the correct execution of the mechanism. To address these challenges, we propose a novel anomaly detection strategy that is designed for real-world FL systems. Our approach activates the defense only when potential attacks are detected, and enables the removal of malicious models without affecting the benign ones. Additionally, we incorporate zero-knowledge proofs to ensure the integrity of the proposed defense mechanism. Experimental results demonstrate the effectiveness of our approach in enhancing FL system security against a comprehensive set of adversarial attacks in various ML tasks.

## 1 Introduction

Federated Learning (FL) (McMahan et al., 2017) enables clients to collaboratively train machine learning models without sharing their local data with other parties. Due to its privacy-preserving nature, FL has attracted considerable attention across various domains in real-world applications (Hard et al., 2018; Chen et al., 2019; Ramaswamy et al., 2019; Leroy et al., 2019; Byrd & Polychroniadou, 2020; Chowdhury et al., 2022). Even though FL does not require clients to share their raw data with other parties, its collaborative nature inadvertently introduces privacy and security vulnerabilities (Cao & Gong, 2022; Bhagoji et al., 2019; Lam et al., 2021; Jin et al., 2021; Tomsett et al., 2019; Chen et al., 2017; Tolpegin et al., 2020a; Kariyappa et al., 2022; Zhang et al., 2022c). Malicious clients can harm training by submitting corrupted model updates to disrupt global model convergence (Fang et al., 2020; Chen et al., 2017), or by planting backdoors that cause the global model to perform poorly on certain data (Bagdasaryan et al., 2020b;a; Wang et al., 2020).

Existing literature on defenses in FL comes with certain inherent limitations, making them unsuitable for real-world FL systems (Blanchard et al., 2017; Yang et al., 2019; Fung et al., 2020; Pillutla et al., 2022; He et al., 2022; Cao et al., 2022; Karimireddy et al., 2020; Sun et al., 2019; Fu et al., 2019; Ozdayi et al., 2021; Sun et al., 2021). Some strategies require prior knowledge of the number of malicious clients within the FL system (Blanchard et al., 2017), while in practice adversaries would not announce their malicious intentions before attacking. Other defense strategies mitigate impacts of potential malicious client submissions by leveraging methods that inevitably alter the aggregation results, such as re-weighting the local models (Fung et al., 2020), modifying the aggregation function (Pillutla et al., 2022), and removing local models that tend to be poisoned (Blanchard et al., 2017). However, in practical FL systems, attacks happen infrequently. While introducing the aforementioned defenses can mitigate the impact of potential malicious clients, the performance loss caused by the inclusion of them can outweigh the defense gain, as most real-world training cases are benign and these defenses largely compromise the model quality for all benign cases. Moreover, existing defense mechanisms are deployed om FL servers without any verification for their execution. As a result, clients are unable to verify whether the defense mechanism was executed accurately and correctly, leaving them reliant on server's integrity and undermining trust in real-world FL systems.

Figure 1: Overview of the proposed anomaly detection mechanism.

Motivated by these, a successful anomaly detection approach should simultaneously satisfy the following: *i*) *detectability*: it should be capable of detecting potential attacks and responding solely when such threats are likely to occur; *ii*) *identifiability*: if an attack is detected, the strategy should further identify the malicious client models and mitigate (or eliminate) their adversarial impacts without harming the benign ones; and *iii*) *verifiability*: the defensive mechanism should be integrated with a verification mechanism to ensure the correct execution of the defense mechanism, such that clients can trust the FL system without relying solely on the server's goodwill.

Table 1: Comparison among our method and state-of-the-art techniques.

| Attribute/Method | Krum | RFA | Foolsgold | NormClip | Bucketing | Median | TrimMean | Ours |
|---|---|---|---|---|---|---|---|---|
| Detecting the presence of attacks | ✗ | ✗ | ✗ | ✗ | ✗ | ✗ | ✗ | ✓ |
| Removing malicious models | ✓ | ✗ | ✗ | ✗ | ✗ | ✗ | ✗ | ✓ |
| Free from impractical knowledge | ✗ | ✓ | ✓ | ✓ | ✓ | ✓ | ✓ | ✓ |
| Free from reweighting | ✓ | ✓ | ✗ | ✓ | ✗ | ✓ | ✓ | ✓ |
| Free from modifying aggregation | ✓ | ✗ | ✓ | ✓ | ✓ | ✗ | ✓ | ✓ |
| Free from harming benign models | ✗ | ✗ | ✗ | ✗ | ✗ | ✓ | ✗ | ✓ |
| *Decent* results in non-attack cases | ✗ | ✗ | ✗ | ✗ | ✗ | ✗ | ✗ | ✓ |
| Verification for correct execution | ✗ | ✗ | ✗ | ✗ | ✗ | ✗ | ✗ | ✓ |

This paper proposes a two-stage defense for anomaly detection that filters out malicious client models in each FL training round with challenges in real-world FL systems addressed. On the first stage, our approach detects potential existence of malicious clients in the current FL round based on *cross-round detection*. The potential presence of malicious clients activates the second stage, named *cross-client detection* that evaluates the *degree of evilness* of each local model and filters out malicious ones based on the intuition of *3σ Rule* (Pukelsheim, 1994). Our mechanism integrates a robust verification protocol that utilizes Zero-Knowledge Proof (ZKP) (Goldwasser et al., 1989) to guarantee integrity and honest execution of the proposed defensive mechanism on the FL server. We overview our mechanism in Figure 1. Then, we compare our approach with state-of-the-art ones, including Krum (Blanchard et al., 2017), RFA (Pillutla et al., 2022), Foolsgold (Fung et al., 2020), NormClip (Sun et al., 2019), Bucketing (Karimireddy et al., 2020), Coordinate-Wise Median (Yin et al., 2018), and Trimmed Mean (Yin et al., 2018) in Table 1. Our contributions are listed below:

*i*) **Real-world applicability.** Our method is designed to meet practical requirements of defenses in real-world FL applications. As far as we know, we are the first to close the significant gap between theoretical research and its real-world applicability in FL security.

*ii*) **Utility and practicability.** Our method is free from any unrealistic prior information, nevertheless it can still detect and eliminate the impact of malicious client models without harming the benign ones. By this means, our method proves it applicability and effectiveness in real-world FL systems where attacks happen rather rarely.

*iii*) **Conditional activation.** We propose a two-stage detection method that first identifies suspicious models and then, if necessary, triggers a double-check of the local models, thereby avoiding unnecessary accuracy loss caused by introducing a defense mechanism.

*iv*) **Accuracy preservation.** Our method preserves accuracy in attack-free situations, which is essential due to the infrequent occurrence of attacks in real-world scenarios.

*v*) **Identifiability.** Our approach removes malicious local models with high accuracy without harming the benign models or modifying the aggregation function.

*vi*) **Verifiability.** To foster trust in FL systems, we leverage ZKPs, enabling clients to independently verify the correct execution of the proposed defense mechanism on the server without relying solely on the server's goodwill.

## 2 PROBLEM SETTING

### 2.1 ADVERSARY MODEL

We consider an FL system in which at least 50% of the clients are benign. Some clients may be adversarial and can conduct attacks to achieve *malicious goals* such as *i)* planting a backdoor so that the global model misclassifies a specific set of samples while the overall model performance is minimally impacted (backdoor attacks, *e.g.*, (Bagdasaryan et al., 2020b; Wang et al., 2020)); *ii)* altering local models to prevent the global model from converging (Byzantine attacks, *e.g.*, (Chen et al., 2017; Fang et al., 2020)); and *iii)* cheating the FL server by randomly submitting contrived models without actual training (free riders, *e.g.*, (Wang, 2022)). We assume the FL server is not fully trusted due to the complex execution environment in real-world FL systems. We assume the FL clients know the server would conduct a defense but they are suspicious if the server has conducted the defense correctly, and they would like to verify the integrity of the defense without depending solely on the server's goodwill. We assume that the adversaries cannot conduct adaptive attacks, and discuss the extension of our approach to adaptive attacks in §3.4.

### 2.2 PRELIMINARIES

**Federated Learning (FL).** FL (McMahan et al., 2017) enables training models across decentralized devices without centralizing data. It is beneficial when dealing with sensitive data, as it allows data to remain on its original device during training.

**Krum.** Krum or $m$-Krum (Blanchard et al., 2017) selects one or $m$ local models that deviate less (evaluated using pairwise distances) from the majority for aggregation. See Appendix A.1 for details.

**3$\sigma$ Rule.** 3$\sigma$ (Pukelsheim, 1994) is an empirical rule and has been utilized in anomaly detection for data management (Han et al., 2019). It states that the percentages of values within one, two, and three standard deviations of the mean are 68%, 95%, and 99.7%, respectively. This rule can be widely applied on real-world applications, as normal distributions are consistent with real-world data distributions (Lyon, 2014). Moreover, when data is not normally distributed, we can transform the distribution to normal distribution (Aoki, 1950; Osborne, 2010; Sakia, 1992; Weisberg, 2001).

**Zero-Knowledge Proofs (ZKPs).** A ZKP (Goldwasser et al., 1989) is a proof system enabling a prover to convince a verifier that a function has been correctly computed on the prover's secret input (witness). ZKPs have three properties: *i)* *correctness*: the proof they produce should pass verification if the prover is honest (integrity property); *ii)* *soundness*: a cheating prover cannot convince the verifier with overwhelming probability, and *iii)* *zero-knowledge*: the prover's witness is not learned by the verifier (privacy property).

## 3 TWO-STAGE ANOMALY DETECTION MECHANISM

We propose a two-stage anomaly detection mechanism to identify and filter out malicious local models on the server. This mechanism is executed at each FL round after the server collects local models from the clients. The server first performs a *cross-round check* that leverages some cache, which we call *reference models*, to assess the *likelihood* of the presence of any malicious clients. Note that at this stage, the server does not remove any local models. If potentially malicious clients are detected, the server subsequently conducts a *cross-client detection* to *analyze* each local model and assess its *degree of evilness*. Based on this evaluation, the server identifies and excludes the malicious models from aggregation.

### 3.1 CROSS-ROUND DETECTION

To assess the likelihood of potential presence of malicious clients, FL servers compute similarities between the local models of the current FL round and certain *golden truth* reference models cached in the last FL round. Local models with higher similarities to the reference models are less likely to be malicious, thus have a higher likelihood to be benign.

We present the intuitive idea in Figure 2. Inspired by the state-of-the-art (Fung et al., 2020), we utilize the cosine score to compute model similarities. For each local model $\mathbf{w}_i$, and its reference model $\mathbf{w}_r$,

---

**Algorithm 1:** Cross-Round Detection

---

**Inputs:** $\tau$: training round id, *e.g.*, $\tau = 0, 1, 2, \ldots$; $\mathcal{W}^{(\tau)}$: client models of $\tau$ round; $\gamma$: upper bound of similarities for malicious client models.

**function** $cross\_round\_check(\mathcal{W}^{(\tau)}, \tau, \gamma)$ **begin**

1    **if** $\tau=0$ **then return** $True$;

2    $\mathcal{W}^{\tau-1} \leftarrow get\_cached\_client\_models()$, $\mathbf{w}_g^{\tau-1} \leftarrow get\_global\_model\_of\_last\_round()$

3    **for** $\mathbf{w}_i^{(\tau)} \in \mathcal{W}^\tau$ **do**

4      $S_c(\mathbf{w}_i^{\tau-1}, \mathbf{w}_i^\tau) \leftarrow get\_similarity(\mathbf{w}_i^{\tau-1}, \mathbf{w}_i^\tau)$, $S_c(\mathbf{w}_g^{\tau-1}, \mathbf{w}_i^\tau) \leftarrow get\_similarity(\mathbf{w}_g^{\tau-1}, \mathbf{w}_i^\tau)$

5      **if** $S_c(\mathbf{w}_g^{\tau-1}, \mathbf{w}_i^\tau) < \gamma$ *or* $S_c(\mathbf{w}_i^{\tau-1}, \mathbf{w}_i^\tau) < \gamma$ **then return** $True$    ▷ There may be attacks ;

6    **return** $False$    ▷ No attack.

---

the cosine similarity is computed as $S_c(\mathbf{w}_i, \mathbf{w}_r) = \frac{\mathbf{w}_i \cdot \mathbf{w}_r}{||\mathbf{w}_i|| \cdot ||\mathbf{w}_r||}$. We expect the cosine similarity of each local model and its reference model to be high, since a higher cosine similarity indicates that the local model is *closer* to the golden truth reference model and, thus, is more likely to be benign. On the contrary, lower cosine similarities indicate that attacks have a higher possibility of occurrence on that client in the current FL training round, as malicious clients may submit arbitrary or tampered local models through some attacks (Bagdasaryan et al., 2020b; Wang et al., 2020; Chen et al., 2017; Fang et al., 2020), making their local models diverge from the reference model.

We select reference models based on the characteristics of the attacks that are widely considered in both the literature and real-world systems, *i.e.*, Byzantine attacks (Chen et al., 2017; Fang et al., 2020) and backdoor attacks (Bagdasaryan et al., 2020b; Wang et al., 2020). For each local model in the current FL training round, we utilize two types of models as the reference models: *i*) the global model from the previous FL training round to identify whether

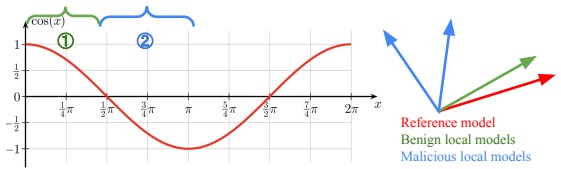

Figure 2: Cosine similarities. ① indicates likely benign models with high cosine similarity, and ② indicates likely malicious models with low cosine similarity.

the current local model deviates significantly from it, potentially preventing the global model from achieving convergence, and *ii*) the local model of the same client from the last FL training round to detect whether the local models submitted by the same client differ much across subsequent rounds, which can indicate that the client was benign in the previous round but turned *evil* in the current round. We note that although we use cosine similarity to compute a bound, our method does not rely heavily on it. In this stage, our method flags suspicious models as *potentially malicious* but does not remove them. Instead, it decides whether to remove them in the latter stage of the proposed approach.

**Cross-Round Detection Algorithm.** We present the cross-round detection algorithm in Algorithm 1. Initially, the server loads the reference models, including the global model from the last FL round, as well as the cached local models that are deemed as *benign* from the previous FL round. For each FL round $\tau$, we denote the global model of the previous FL round by $\mathbf{w}_g^{\tau-1}$. We let $\mathbf{w}_i^\tau$ denote local model submitted by client $\mathcal{C}_i$ in the current round $\tau$, and let $\mathbf{w}_i^{\tau-1}$ denote that client's cached local model from the previous round. The algorithm computes similarities $S_c(\mathbf{w}_i^\tau, \mathbf{w}_g^{\tau-1})$ and $S_c(\mathbf{w}_i^\tau, \mathbf{w}_i^{\tau-1})$, and utilizes these scores, together with a threshold $\gamma$ ($-1 < \gamma < 1$), to detect whether potential attacks have happened in the current FL training round. Any similarity score that is lower than $\gamma$ signals that the corresponding client might be malicious and triggers a further inspection on the client models in the second stage of our anomaly detection approach, as described in §3.2.

## 3.2 CROSS-CLIENT DETECTION

Cross-client detection computes a score for each local model to evaluate its *degree of evilness*, and utilizes the $3\sigma$ rule to filter out those local models with higher degrees of evilness, *i.e.*, the malicious models. The $3\sigma$ rule is pivotal for three reasons: *i*) in case the client datasets are i.i.d., parameters of the local models follow normal distribution (Baruch et al., 2019; Chen et al., 2017; Yin et al., 2018); *ii*) according to the Central Limit Theorem (CLT) (Rosenblatt, 1956), when client datasets are

---

**Algorithm 2:** Cross-Client Detection Algorithm.

---

**Inputs:** $\tau$: training round id, $\tau = 0, 1, 2, \ldots$; $\mathcal{W}$: local models of a training round; $m$: $m$-Krum parameter.

**function** $Cross\_Client\_Detection(\mathcal{W}, \tau)$ **begin**

  1  |  **if** $\tau = 0$ **then**
      |    $\lfloor$  $m \leftarrow |\mathcal{W}|/2, f \leftarrow |\mathcal{W}|/2, \mathbf{w}_{\text{avg}} \leftarrow Krum\_and\_m\_Krum(\mathcal{W}, m, f)$

  2  |  $\mathcal{L} \leftarrow compute\_L2\_scores(\mathcal{W}, \mathbf{w}_{\text{avg}})$

  3  |  $\mu \leftarrow \frac{\sum_{\ell \in \mathcal{L}} \ell}{|\mathcal{L}|}, \sigma \leftarrow \sqrt{\frac{\sum_{\ell \in \mathcal{L}} (\ell - \mu)^2}{|\mathcal{L}| - 1}}$    $\triangleright$ Compute $\mathcal{N}(\mu, \sigma)$

  4  |  **for** $0 < i < |\mathcal{W}|$ **do**

  5  |    $\lfloor$  **if** $\mathcal{L}[i] > \mu + \lambda\sigma$ **then** remove $\mathbf{w}_i$ from $\mathcal{W}$ ;

  6  |  $\mathbf{w}_{\text{avg}} \leftarrow average(\mathcal{W})$    $\triangleright$ Cache $\mathbf{w}_{\text{avg}}$

  7  |  **return** $\mathcal{W}$

---

non-i.i.d., the local models still converge towards normal distribution, especially when the number of clients is at least 30 (Chang et al., 2006; of Public Health, 2001); and *iii*) even when CLT does not hold strongly (*e.g.*, the number of clients is lower than 30), previous works show that the local models still exhibit certain statistical features (Karimireddy et al., 2020; Pillutla et al., 2022), thus the $3\sigma$ rule can still be applied to derive analytics from the local models.

Let $\mathcal{L}$ denote the *degree of evilness* for client models in the current FL round, where higher scores indicate higher probability for that client to be malicious. Suppose $\mathcal{L}$ follows normal distribution $\mathcal{N}(\mu, \sigma)$, where $\mu$ is mean and $\sigma$ is standard deviation. We then have the following definition.

**Definition 3.1.** Local models with *evilness degree* higher than $\mu + \lambda\sigma$ are identified as malicious local models, where $\lambda$ ($\lambda > 0$) adjusts the sensitivity of the score computation.

According to Definition 3.1, local models with *degree of evilness* higher than the boundary are detected as *malicious models* and are excluded from aggregation. We note that we only take one side of the bounds given by the $3\sigma$ rule, such that the models with evilness lower than $\mu + \lambda\sigma$ are not identified as outliers since we prefer lower evilness.

The details are described in Algorithm 2. In this paper, we select $L_2$ distances to compute the *degree of evilness*. For each local model, we compute its score using that model and the average model from the previous round, denoted as $\mathbf{w}_{\text{avg}}$. We prefer that the local model does not deviate significantly from the average model of the previous round, which can serve as *golden truth*. For each local model $\mathbf{w}_i$ in the current round, its $L_2$ score, denoted as $\mathcal{L}[i]$, is computed as $\mathcal{L}[i] = ||\mathbf{w}_i - \mathbf{w}_{\text{avg}}||$. Considering that the first round does not have an average model as a reference, to avoid involving any malicious models in the aggregation of the first round, we utilize $m$-Krum (Blanchard et al., 2017) to compute an approximate average model. In $m$-Krum, it is ideal to involve a maximum number of benign local models and avoid polluting the approximate average model from any malicious local model. As the FL server does not know the number of potential malicious clients, we set $m$ to $|\mathcal{W}|/2$ to compute an approximate average model based on the assumption that the number of malicious clients is less than $|\mathcal{W}|/2$, where $|\mathcal{W}|$ is the number of clients in each FL round. In later training rounds, we do not need $m$-Krum as we simply utilize the average model from the previous round.

### 3.3 OPTIMIZATIONS FOR REFERENCE MODELS

So far, the server stores the complete client models and the updated global model as reference models for the next FL round at the end of each FL round. However, this approach encounters pragmatic challenges in real-world deployments due to the following: *i*) *Storage Constraints:* real-world FL systems often have complex execution environments and restricted storage, which necessitate the algorithm to be optimized for storage and computation efficiency; *ii*) *Computational Overhead:* incorporating a ZKP for validation after each FL round (which will be discussed in §4) is computationally intensive (Goldreich & Krawczyk, 1996). Utilizing the entire collection of client/global models for computation increases resource consumption significantly and prolongs the verification time in each FL round. Meanwhile, the FL system must await the completion of this verification process before continuing the subsequent operations, which detrimentally impacts the experience of the FL clients.

Figure 3: ZKP circuits for the proposed two-stage anomaly detection mechanism.

In light of these, we propose using only segmental models instead of entire models as reference models. The reference model should follow the following criteria: *i*) the selected fragment should sufficiently represent the full model while minimizing the fragment size, ideally using just one layer of the original model; *ii*) the selection mechanism must be generally applicable in real-world systems and independent of specific data distributions or model structures. We follow the terminology in Fung et al. (2020) and name such layer as an *importance layer*. We note that such a layer is not required to contain the maximal information compared to other layers of the same model, but should be more *informative* than the majority of the other layers. Intuitively, we select the second-to-last layer as the importance layer, as it is close to the output layer and thus can retain substantial information. This method can reduce complexity effectively, especially for ZKP-related computations. As an example, the second-to-last layer of CNN contains only $7,936$ parameters, compared to its full size of $1,199,882$ parameters. We experimentally validate our importance layer selection in **Exp 1** in §5.

### 3.4 DISCUSSIONS ON EXTENSIONS TO ADAPTIVE ATTACKS

In this work, we focus on non-adaptive adversaries, a common assumption in the literature (Ozdayi et al., 2021; Guerraoui et al., 2018b; Pillutla et al., 2022; Karimireddy et al., 2020; Yin et al., 2018) that enables us to create a baseline for anomaly detection and establish the robustness of our model under basic adversarial settings. While adaptive adversaries present a more challenging scenario, addressing them would introduce complexities that are beyond the scope of this initial work.

We acknowledge the importance of addressing adaptive attacks in FL systems. Below we discuss extensions of our approach to adaptive attacks. In the presence of adaptive attackers, malicious clients can craft their models based on their knowledge of the global model, their local model from the last FL iteration, and the cosine similarity threshold used in our defense. By carefully modifying their local models to ensure the cosine similarity falls within the threshold, malicious models may appear benign and survive the detection. To solve this problem, our approach can be extended from the following two directions: 1) *regularizing local models before measuring cosine similarity*, such that the adversaries cannot craft their local models based on their known information; and 2) *diversifying the layers used for cross-round detection* instead of just relying on a single layer, thus it would be difficult for adversaries to predict and modify their local models accordingly.

## 4 VERIFIABLE ANOMALY DETECTION

Our method incorporates a verification module to enhance trust and privacy within the FL system. Ideally, the verification module should have the following features: *i*) *client-side verification*: it should enable clients who may have concerns about the integrity of the FL server to verify the correct execution of the defense mechanism, without solely relying on the server's goodwill; and *ii*) *privacy protection*: it should not necessitate the clients to access inappropriate knowledge, such as local models from other clients, thus preserving privacy and integrity in the FL system.

We incorporate ZKPs to ensure that the clients can verify the integrity of the defense without accessing the other local models. We utilize zkSNARKs (Bitansky et al., 2012) that offer constant proof sizes and constant verification time regardless of the size of computation. Such property is crucial for applications where the verifier's (*i.e.*, an FL client) resources are limited, *e.g.*, real-world FL systems. We design ZKP circuits as in Figure 3. Details of implementations are in Appendix §A.4.

**ZKP for Anomaly Detection.** Most of the computations in Algorithm 1 and Algorithm 2 are linear and can be compiled into an arithmetic circuit easily, *e.g.*, computing cosine similarity between

two matrices of size $n \times n$ requires a circuit with $O(n^2)$ multiplication gates and one division. While it is difficult to directly compute division on a circuit, it can be easily verified with the prover providing the pre-computed quotient and remainder beforehand. Similar to Weng et al. (2021), we can utilize Freivalds' algorithm (Freivalds, 1977) to verify matrix multiplications. In general, the matrix multiplication constitutes the basis of the verification schemes used for the proposed mechanism. Naively verifying a matrix multiplication $AB = C$ where $A, B, C$ are of size $n \times n$ requires proving the computation step by step, which requires $O(n^3)$ multiplication gates. With Freivalds' algorithm, the prover first computes the result off-circuit and commits to it. Then, the verifier generates a random vector $v$ of length $n$, and checks $A(Bv) \stackrel{?}{=} Cv$. This approach reduces the size of the circuit to $O(n^2)$. We exploit this idea to design an efficient protocol for the square root computation in Algorithm 2. To verify that $x = \sqrt{y}$ is computed correctly, we ask the prover to provide the answer $x$ as witness and then we check in the ZKP that $x$ is indeed the square root of $y$. Note that we cannot check $x^2$ is equal to $y$ because the zkSNARK works over a prime field and the square root of an input number might not exist. So, we check if $x^2$ is close to $y$ by checking that $x^2 \leq y$ and $(x+1)^2 \geq y$. This approach reduces the computation of square root to 2 multiplications and 2 comparisons.

The zero-knowledge property of ZKPs allows public verification of prover's (*i.e.*, the FL server) integrity in case of the server being untrustworthy. By incorporating ZKPs, we provide a public verifiable approach for each client to ensure FL server's integrity which is essential for building and maintaining trust in FL systems. This ensures that clients can verify the correctness of the defense without needing to rely solely on the server's goodwill. This is also secure in case there exists adversarial clients, as the ZKP itself reveals nothing about the prover's witness, i.e., private data, models, and/or thresholds the server uses during the proposed anomaly detection approach.

## 5 EVALUATIONS

**Setting.** A summary of datasets and models for evaluations can be found in Table 2. By default, we employ CNN and the non-i.i.d. FEMNIST dataset ($\alpha = 0.5$), as the non-i.i.d. setting closely captures real-world scenarios. We utilize FedAVG in our experiments. By default, we use 10 clients for FL training, corresponding to real-world FL applications where the number of clients is typically less than 10, especially in ToB scenarios. We also vary the number of clients from 10 to 100 in **Exp 5**, and validate the utility of our approach in a practical application using 20 edge real-world devices; see **Exp 11**. We conduct our evaluations on a server with 8 NVIDIA A100-SXM4-80GB GPUs, and validate the correct execution with ZKP on Amazon AWS with an m5a.4xlarge instance with 16 CPU cores and 32 GB memory. We implement the ZKP system in Circom (Contributors, 2022).[1]

Table 2: Models and datasets.

| Model | Dataset |
|---|---|
| CNN (McMahan et al., 2017) | FEMNIST (Caldas et al., 2018) |
| ResNet-20 (He et al., 2016) | Cifar10 (Krizhevsky et al., 2009) |
| ResNet-56 (He et al., 2016) | Cifar100 (Krizhevsky et al., 2009) |
| RNN (McMahan et al., 2017) | Shakespeare (McMahan et al., 2017) |
| LR (Cox, 1958) | MNIST (Deng, 2012) |

**Selection of attacks and defenses.** We employ two byzantine attacks and two backdoor attacks that are widely considered in literature, including a random weight Byzantine attack that randomly modifies the local submissions (Chen et al., 2017; Fang et al., 2020), a zero weight Byzantine attack that sets all model weights to zero (Chen et al., 2017; Fang et al., 2020), the label flipping backdoor attack that flip labels in the local data Tolpegin et al. (2020b), and a model replacement backdoor attack (Bagdasaryan et al., 2020b) that intends to use a poisoned local model to replace the global model. We utilize 5 baseline defense mechanisms that can be effective in real systems: $m$-Krum (Blanchard et al., 2017), Foolsgold (Fung et al., 2020), RFA (Pillutla et al., 2022), Bucketing (Karimireddy et al., 2020), and Trimmed Mean (Yin et al., 2018). For $m$-Krum, we set $m$ to 5, which means 5 out of 10 submitted local models participate in aggregation in each FL training round.

**Evaluation Metrics.** We evaluate the effectiveness of cross-round check using *cross-round detection success rate*, defined by the proportion of rounds where the algorithm correctly detects cases with

---

[1]We provide a link to our code in Appendix §A.7.

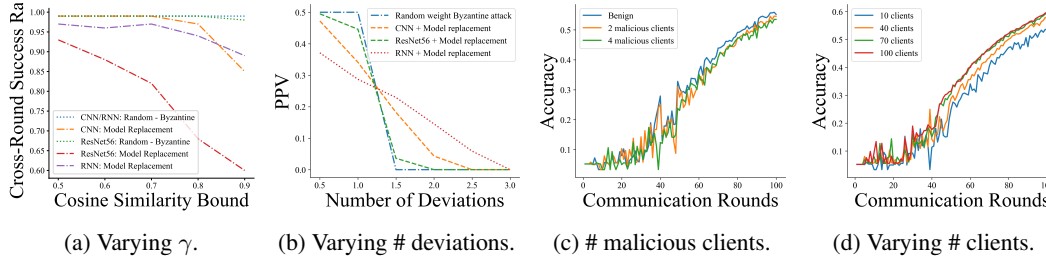

(a) Varying $\gamma$.     (b) Varying # deviations.     (c) # malicious clients.     (d) Varying # clients.

Figure 4: Impacts of different parameters.

or without an attack relative to the number of total FL rounds. A 100% cross-round success rate indicates that all FL rounds that potential attacks might have happened are detected, and none of the benign cases are identified as "attacks" by mistake. We evaluate the quality of cross-client detection using modified Positive Predictive Values (PPV) (Fletcher, 2019), the proportions of positive results in statistics and diagnostic tests that are true positive results. Let us denote the number of true positive and the false positive results as $N_{TP}$ and $N_{FP}$, respectively. Then we have $PPV = \frac{N_{TP}}{N_{TP}+N_{FP}}$. In our setting, client submissions that are detected as "malicious" and are actually malicious are defined as *True Positive*, *i.e.*, $N_{TP}$, while client submissions that are detected as "malicious" even though they are benign are defined as *False Positive*, *i.e.*, $N_{FP}$. Since we would like the PPV to reveal the relation between $N_{TP}$ and the total number of malicious local models across all FL rounds, we use the total number of malicious local models across all FL rounds, denoted as $N_{total}$, and compute a modified PPV as $\frac{N_{TP}}{N_{TP}+N_{FP}+N_{total}}$, where $0 \leq PPV \leq \frac{1}{2}$. Ideally, PPV is $\frac{1}{2}$, where all malicious local models are detected, *i.e.*, $N_{FP} = 0$ and $N_{TP} = N_{total}$. The details are in Appendix A.2.

**Exp 1: Selection of importance layer.** We utilize the L2-norm of the local models to evaluate the "sensitivity" of each layer. A layer with a norm higher than most of the other layers indicates higher sensitivity compared to others, thus can be utilized to represent the whole model. We evaluate the sensitivity of the layers of CNN, RNN, and ResNet-56. The results for RNN, CNN, and ResNet-56 are deferred to Figure 9a, Figure 9b, and Figure 9c in Appendix §A.5, respectively. The results show the sensitivity of the second-to-the-last layer is higher than most of the other layers. Thus, this layer includes adequate information of the whole model and can be selected as the importance layer.

**Exp 2: Impact of the similarity threshold.** We evaluate the impact of the similarity threshold $\gamma$ in the cross-round check with 10 clients in each FL round, where 4 of them are malicious. Ideally, the cross-round check should confirm the absence or presence of an attack accurately. We evaluate the impact of the cosine similarity threshold $\gamma$ in the cross-round check by setting $\gamma$ to 0.5, 0.6, 0.7, 0.8, and 0.9. As described in Figure 4a, the cross-round detection success rate is close to 100% in the case of Byzantine attacks. We observe that, when the cosine similarity threshold $\gamma$ is set to 0.5, the performance is satisfactory in all cases, with at least 93% cross-round detection success rate.

**Exp 3: Selection of the number of deviations** ($\lambda$). We set $\lambda$ to 0.5, 1, 1.5, 2, 2.5, and 3, and utilize $PPV$ to evaluate the impact of the number of deviations, *i.e.*, the parameter $\lambda$ in the anomaly bound $\mu + \lambda\sigma$. To evaluate a challenging case where a large portion of the clients are malicious, we set 40% clients malicious in each FL round. Given that the number of FL rounds is 100, the total number of malicious submissions $N_{total}$ is 400. We evaluate our approach on three tasks, as follows: *i*) CNN+FEMNIST, *ii*) ResNet-56+Cifar100, and *iii*) RNN + Shakespeare. We observe in Figure 4b, that when $\lambda$ is 0.5, the results are the best. Especially for the random weight Byzantine attack, we see that the $PPV$ is exactly 0.5, indicating that all malicious local models are detected. In subsequent experiments, unless specified otherwise, we set $\lambda$ to 0.5.

**Exp 4: Varying the percentage of malicious clients.** We use random Byzantine attack and set the percentage of malicious clients to 20% and 40%. We also include a baseline case where all clients are benign. As shown in Figure 4c, the test accuracy remains relatively consistent across different cases, as in each FL training round, our approach filters out the local models that tend to be malicious to minimize the negative impact of malicious client models on aggregation.

**Exp 5: Varying the number of FL clients.** We explore the impact of the number of clients under the random Byzantine attack. We set the number of clients to 10, 40, 70, and 100, and set the percentage of malicious clients to 40%. The results, as described in Figure 4d, indicate that in all cases, our approach has high utility and can filter out malicious clients with high accuracy.

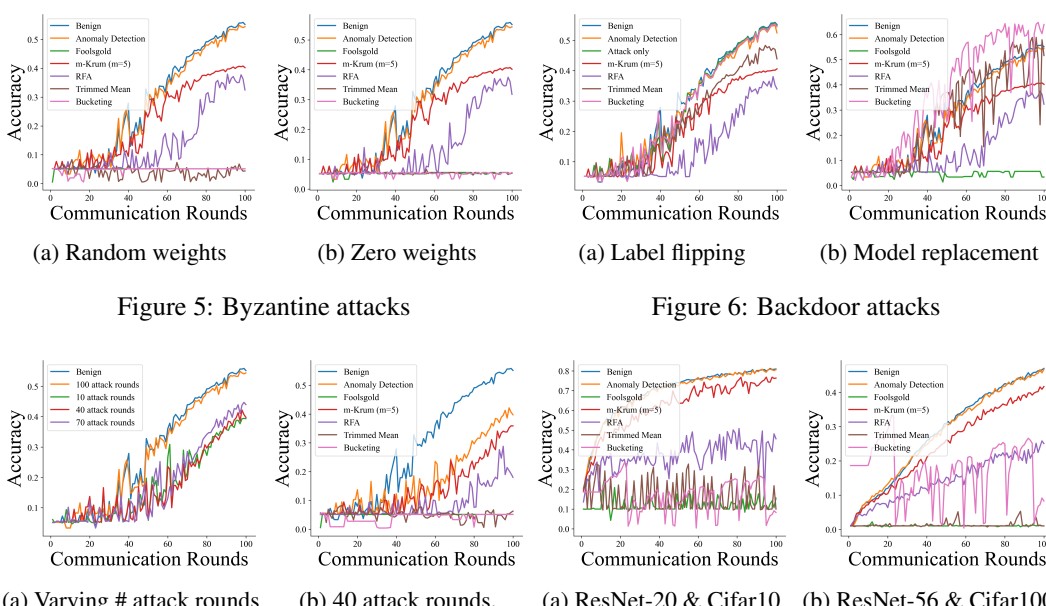

(a) Random weights    (b) Zero weights      (a) Label flipping    (b) Model replacement

Figure 5: Byzantine attacks        Figure 6: Backdoor attacks

(a) Varying # attack rounds   (b) 40 attack rounds.    (a) ResNet-20 & Cifar10   (b) ResNet-56 & Cifar100

Figure 7: Evaluations on selected attacks      Figure 8: Evaluations on CV tasks

**Exp 6: Evaluations on Byzantine attacks.** We compare our approach with the state-of-the-art defenses using 10 clients, and set one of them as malicious in each FL round. We include a "benign" baseline scenario with no activated attack or defense. The results for the random weight Byzantine attack (Figure 5a) and the zero weight Byzantine attack (Figure 5b) demonstrate that our approach effectively mitigates the negative impact of the attacks and significantly outperforms the other defenses, by achieving a test accuracy much closer to the benign case.

**Exp 7: Evaluations on backdoor attacks.** We compare our approach with the state-of-the-art defenses using 10 clients, where one of them is malicious in each FL round. Considering that the label flipping attack is subtle and manipulates local training data and produces malicious local models that are challenging to detect, we set the parameter $\lambda$ to 2 to produce a tighter boundary. The results for the label flipping attack and model replacement backdoor attack are shown in Figure 6a and Figure 6b, respectively. Results show that our approach is effective against backdoor attacks, with the test accuracy much closer to the benign case compared to the baseline defenses.

**Exp 8: Evaluations on different attack frequencies.** We configure attacks to occur only during specific rounds to evaluate the effectiveness of the proposed two-stage approach. The total number of attack rounds is set to 10, 40, 70, and 100, respectively. We then fix the number of attack rounds to 40 and compare our approach with the state-of-the-art defenses. The results in Figure 7a and Figure 7b show that our method effectively mitigates the impact of the adversarial attacks, ensuring minimal accuracy loss and robust performance even under different attack rounds.

**Exp 9: Evaluations on different tasks.** We evaluate the defenses against the random mode of the Byzantine attack with different models and datasets described in Table 2. The results in Figure 8a, Figure 8b, and Figure 9d in §A.5 show that our approach outperforms the baseline defenses by effectively filtering out poisoned local models, with a test accuracy close to the benign scenarios. Moreover, some defenses may fail in some tasks, *e.g.*, $m$-Krum fails in RNN in Figure 9d, as those methods either select a fixed number of local models or re-weight the local models in aggregation, which potentially eliminates some local models that are important to the aggregation, leading to an unchanged test accuracy in later FL rounds.

**Exp 10: Evaluations of ZKP verification.** We implement a prover's module which contains JavaScript code to generate witness for the ZKP, as well as to perform fixed-point quantization. Specifically, we only pull out parameters of the importance layer to represent the whole model to reduce complexity. We report the results in Table 3.

**Exp 11: Evaluations in a real-world setting.** To validate the utility and scalability of our approach in real-world applications, we utilize 20 real-world edge devices to demonstrate how our anomaly

Table 3: Cost of ZKP of different models

| Model | Stage 1 Circuit Size | Stage 2 Circuit Size | Proving Time (s) | Verification Time (ms) |
|-------|---------------------|---------------------|------------------|------------------------|
| CNN | 476,160 | 795,941 | 33 (12 + 21) | 3 |
| RNN | 1,382,400 | 2,306,341 | 96 (34 + 62) | 3 |
| ResNet-56 | 1,536,000 | 2,562,340 | 100 (37 + 63) | 3 |

Bracketed times denote duration for cross-round detection and cross-client detection.

detection mechanism performs under practical constraints and settings. The device information is shown in Figure 10 in Appendix §A.6. In each FL round, we designate 5 devices as malicious. The FL client package is integrated into the edge nodes to fetch data from our back-end periodically. Due to the challenges posed by real-world settings, such as devices equipped solely with CPUs (lacking GPUs), potential connectivity issues, network latency, and limited storage on edge devices, we select a simple task, i.e., using the MNIST dataset for a logistic regression task, and use our proposed anomaly detection method to prevent against the random weight Byzantine attack. The training process is shown in Figure 11 in Appendix §A.6 , and the total training time is 221 seconds. The CPU utilization and network traffic during training are shown in Figure 12 and Figure 13 in Appendix §A.6, respectively.

## 6 RELATED WORKS

**Detection of attacks.** Zhang et al. (2022b) employs $k$-means to partition local models into clusters that correspond to "benign" or "malicious". While this approach can efficiently detect attacks, it requires some pre-training rounds and relies much on historical client models, thus might not be as effective when there is limited information on past client models. For example, their implementation (Zhang et al., 2022a) sets the starting round to detect attacks to different training rounds, *e.g*., 50 when the datasets are MNIST and FEMNIST, and 20 when the dataset is CIFAR10. While this approach is novel, it is not suitable for real FL systems, as attacks may happen in earlier rounds as well.

**Defense mechanisms in FL.** Robust learning and the mitigation of adversarial behaviors in FL has been extensively explored (Blanchard et al., 2017; Yang et al., 2019; Fung et al., 2020; Pillutla et al., 2022; He et al., 2022; Karimireddy et al., 2020; Sun et al., 2019; Fu et al., 2019; Ozdayi et al., 2021; Sun et al., 2021; Yin et al., 2018; Chen et al., 2017; Guerraoui et al., 2018a; Xie et al., 2020; Li et al., 2020; Cao et al., 2020). Some approaches keep several local models that are more likely to be benign in each FL round, *e.g*., (Blanchard et al., 2017; Guerraoui et al., 2018a; Yin et al., 2018), and (Xie et al., 2020), instead of aggregating all client submissions. Such approaches are effective, but they keep fewer local models than the real number of benign local models to ensure that all malicious local models are filtered out, causing misrepresentation of some benign local models in the aggregation. This completely wastes the computation resources of the benign clients that are incorrectly removed and thus, changes the aggregation results. Some approaches re-weight or modify local models to mitigate the impacts of potential malicious submissions (Fung et al., 2020; Karimireddy et al., 2020; Sun et al., 2019; Fu et al., 2019; Ozdayi et al., 2021; Sun et al., 2021), while other approaches alter the aggregation function or directly modify the aggregation results (Pillutla et al., 2022; Karimireddy et al., 2020; Yin et al., 2018; Chen et al., 2017). While these defense mechanisms can be effective against attacks, they might inadvertently degrade the quality of outcomes due to the unintentional alteration of aggregation results even when no attacks are present. This is especially problematic given the low frequency of attacks in practical scenarios.

## 7 CONCLUSIONS

We present a novel anomaly detection approach specifically designed for real-world FL systems. Our approach utilizes an early cross-round check that activates subsequent anomaly detection exclusively in the presence of attacks. When attacks happen, our approach removes malicious client models efficiently, ensuring that the local models submitted by benign clients remain unaffected. By leveraging ZKPs, our approach enables clients to verify the integrity of the removal performed by the server.

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

# A APPENDIX

## A.1 DETAILS OF KRUM AND $m$-KRUM

In Krum and $m$-Krum, the server selects $m$ ($m$ is one in Krum) local models that deviate less from the majority based on their pairwise distances, where such local models are more likely to be benign and thus are accepted for aggregation in the current round. Given that there are $f$ byzantine clients among $L$ clients that participate in each FL iteration, Krum selects one model that is the most likely to be benign as the global model. That is, instead of using all $L$ local models in aggregation, the server selects a single model to represent all $L$ submissions. To do so, Krum computes a score for each model $\mathbf{w}_i$, denoted as $S_K(\mathbf{w}_i)$, using $L - f - 2$ local models that are "closest" to $\mathbf{w}_i$, and selects the local model with the minimum score to represent the aggregation result. For each local model $\mathbf{w}_i$, suppose $C_i^{\mathcal{N}}$ is the set of the $L - f - 2$ local models that are closest to $\mathbf{w}_i$, then $S_K(\mathbf{w}_i)$ is computed by

$$S_K(\mathbf{w}_i) = \sum_{j \in \mathcal{C}_i} ||\mathbf{w}_i - \mathbf{w}_j||^2.$$

An optimization of Krum is $m$-Krum (Blanchard et al., 2017) that selects $m$ local models, instead of one, when aggregating local models. The algorithm for Krum and $m$-Krum is summarized in Algorithm 3 .

## A.2 PROOF OF THE RANGE OF PPV

Below, we show that the upper bound of $PPV$ is $\frac{1}{2}$.

*Proof.* $PPV = \frac{N_{TP}}{N_{TP} + N_{FP} + N_{total}}$, then $\frac{1}{PPV} = 1 + \frac{N_{FP}}{N_{TP}} + \frac{N_{total}}{N_{TP}}$. As $\frac{N_{FP}}{N_{TP}} \geq 0$ and $\frac{N_{total}}{N_{TP}} \geq 1$, we have $\frac{1}{PPV} \geq 2$, thus $PPV \leq \frac{1}{2}$. ☐

---

**Algorithm 3:** Krum and $m$-Krum.

---

**Inputs:** $\mathcal{W}$: client submissions of a training round; $i$: the client id for which we compute a Krum score $S_K(\mathbf{w}_i)$; $f$: the number of malicious clients in each FL iteration; $m$: the number of "neighbor" client models that participate in computing the Krum score $S_k(\mathbf{w}_i)$ of each client model $\mathbf{w}_i$; $m$ is 1 by default in Krum.

1 **function** *Krum_and_m_Krum($\mathcal{W}, m, f$)* **begin**
2     $S_k \leftarrow []$
3     **for** $\mathbf{w}_j \in \mathcal{W}$ **do**
4        $S_k(\mathbf{w}_i) \leftarrow compute\_krum\_score(\mathcal{W}, i, m, f)$
5     $filter(\mathcal{W}, S_k)$    $\triangleright$ Keep local models with the $L/2$ lowest Krum scores
6     **return** $average(\mathcal{W})$
7 **function** *compute_krum_score($\mathcal{W}, i, m, f$)* **begin**
8     $d \leftarrow []$    $\triangleright$ Square distances of $\mathbf{w}_i$ to other local models.
9     $L \leftarrow |\mathcal{W}|$    $\triangleright$ $L$: the number of clients in each FL round.
10    **for** $\mathbf{w}_j \in \mathcal{W}$ **do**
11      **if** $i \neq j$ **then**
        $d.append(||\mathbf{w}_i - \mathbf{w}_j||^2)$
12    $sort(d)$    $\triangleright$ In ascending order
13    $S_k(\mathbf{w}_i) \leftarrow \sum_{k=0}^{L-f-3} d$    $\triangleright$ Use the smallest $L - f - 2$ scores to compute $S_k(\mathbf{w}_i)$
14    **return** $S_k(\mathbf{w}_i)$

---

### A.3 EXTENSION TO CLIENT SAMPLING

Our method can work in the case of client sampling. For ease of explanation, in the main manuscript, we assumed that all clients participate in aggregation in every FL iteration. However, with some engineering efforts, we can easily extend the method to handle client selection. To handle scenarios with client selection, we can cache historical client models for the same clients across rounds, such that the server can perform cross-round detection even when clients do not participate in every round. If the cached model for a client is too old, we can use the global model from the last round as the reference model. A scenario with adversary clients that participate only once (*i.e.,* single-shot attacks) constitutes a specific case of the client selection challenge described above. In such cases, we can use the global model from the last round as the reference model for cross-round detection.

### A.4 ZERO-KNOWLEDGE PROOF (ZKP) IMPLEMENTATION

This section describes the details of the implementation of ZKPs. In what follows, the prover is the FL server, whereas the verifiers are the FL clients.

#### A.4.1 CHOICE OF THE ZKP SYSTEM

In our implementation, we use the Groth16 (Groth, 2016) zkSNARK scheme implemented in the Circom library (Contributors, 2022) for all the computations described earlier. We choose this ZKP scheme because its construction ensures constant proof size (128 bytes) and constant verification time. Because of this, Groth16 is popular for blockchain applications as it necessitates little on-chain computation. There are other ZKP schemes based on different constructions that can achieve faster prover time (Liu et al., 2021), but their proof size is bigger and verification time is not constant, which is a problem if the verifier lacks computational power, as in our case since the verifiers are the FL clients in our setting. The construction of a ZKP scheme that is efficient for both the prover and verifier is still an open research direction.

#### A.4.2 ZKP-COMPATIBLE LANGUAGE

The first challenge of applying ZKP protocols is to convert the computations into a ZKP-compatible language. ZKP protocols model computations as arithmetic circuits with addition and multiplication gates over a prime field. However, our computations for our approach are over real numbers. The second challenge is that some computations such as square root are nonlinear, making it difficult to wire them as a circuit. To address these issues, we implement a class of operations that map real

numbers to fixed-point numbers. To build our ZKP scheme, we use Circom library (Contributors, 2022), which compiles the description of an arithmetic circuit in a front-end language similar to C++ to the back-end ZKP protocol.

### A.4.3   INTERACTIVITY OF ZKSNARKS

In the Freivalds' algorithm (Freivalds, 1977), the prover first computes the matrix multiplication and commits to its result. Then the verifier generates and sends the random vector. This step is interactive in nature, but we can make this non-interactive using the Fiat-Shamir heuristic as it is public-coin, meaning the vector is randomly selected by the verifier and made public to everyone. Therefore, the prover can instead generate this vector by setting it to the hash of matrices A,B and C. With this, our entire ZKP pipeline, including the Freivalds' step can become truly non-interactive.

### A.4.4   MOTIVATION OF IMPLEMENTING ZKP

ZKP enables proving to the clients that the server has correctly executed the anomaly detection process. This addresses a critical concern in FL systems, where clients cannot directly verify the server's behavior and must fully trust the server. Below, we explain the motivation for ZK from research, industry product, and system perspectives.

**Research Perspective:**  Existing literature has considered various adversarial models. For example, 1) clients might be malicious and submit modified models; 2) FL server might be curious about local models and want to infer sensitive information, such as original training data, or the local models; 3) clients might be curious about local models of other clients; 4) an external adversary may hack the communication channels between clients and the server and poison some client models; 5) the FL server may be hacked by external adversaries; 6) a global "sybil" may hack the whole system and control some clients by modifying their local training data, and so on.

In our paper, we assume the FL server is not fully trusted due to the complex execution environment in real systems. There may be external adversaries or a global sybil, thus, even if the server hopes to execute the aggregation correctly, the presence of adversaries necessitates a ZKP module for verification to ensure that the server's actions are transparent and trustworthy to all clients.

**Industry Perspective:**  The necessity of ZKP also arises from real-world application needs. Consider, for example, FL clients that are medical institutions or hospitals holding sensitive data, such as patient medical records. These institutions may want to collaboratively train a model but be unwilling to share their raw data due to privacy concerns. Although these institutions know that the server will run an anomaly detection procedure, they may not be fully convinced that the server will honestly execute the procedure or that their models will participate in the aggregation without bias. Here, ZKP enables verification that the anomaly detection is performed correctly, even when the clients do not have access to the local models of other clients. This is critical for gaining the trust of the participating clients.

**System Perspective:**  Real FL systems with rewards contain components such as model aggregation, contribution assessment of local models, and anomaly detection, etc. If the FL server is not fully trusted, validating all these operations is essential. However, the focus of our paper is specifically on anomaly detection, and therefore, we have primarily discussed the application of ZKP in this context. The ZKP module ensures that even if the server is not fully trusted, e.g., under potential external threats, clients can have verifiable proof that the anomaly detection has been executed correctly, thus maintaining the integrity and security of the whole FL process.

### A.5   SUPPLEMENTARY EXPERIMENTAL RESULTS

The results for the importance layers of RNN, CNN, and ResNet-56 are given in Figure 9a, Figure 9b, and Figure 9c, respectively. The results for evaluations on RNN and the Shakespeare dataset is shown in Figure 9d.

### A.6   SUPPLEMENTARY RESULTS FOR THE REAL-WORLD EXPERIMENT

The edge devices we use are described in Figure 10, the real-world simulation is in Figure 11, the CPU utilization is in Figure 12, and the network traffic is in Figure 13.

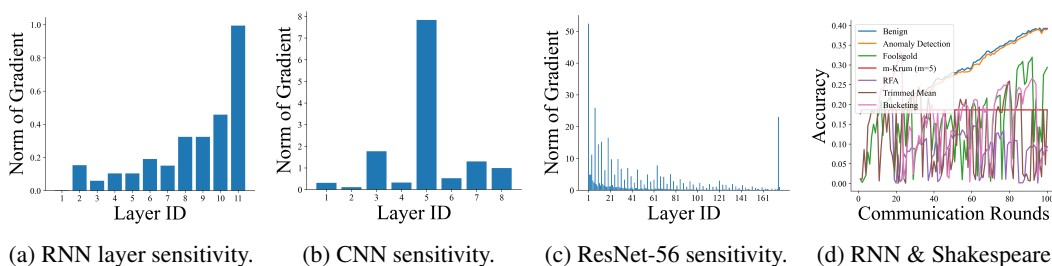

(a) RNN layer sensitivity.    (b) CNN sensitivity.    (c) ResNet-56 sensitivity.    (d) RNN & Shakespeare

Figure 9: Supplementary experimental results.

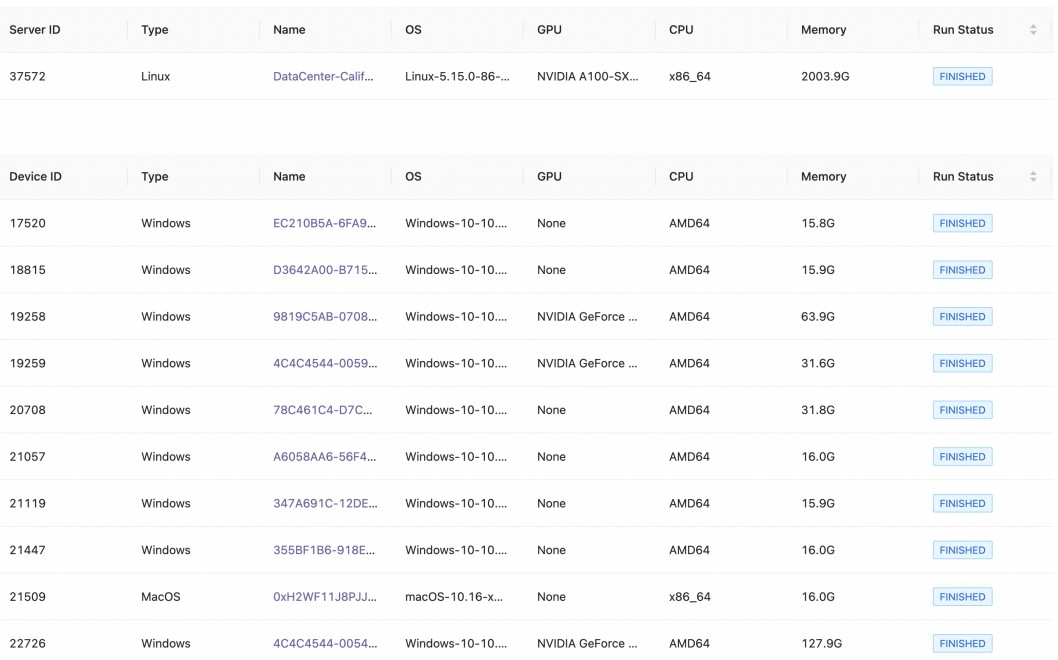

Figure 10: Edge device information.

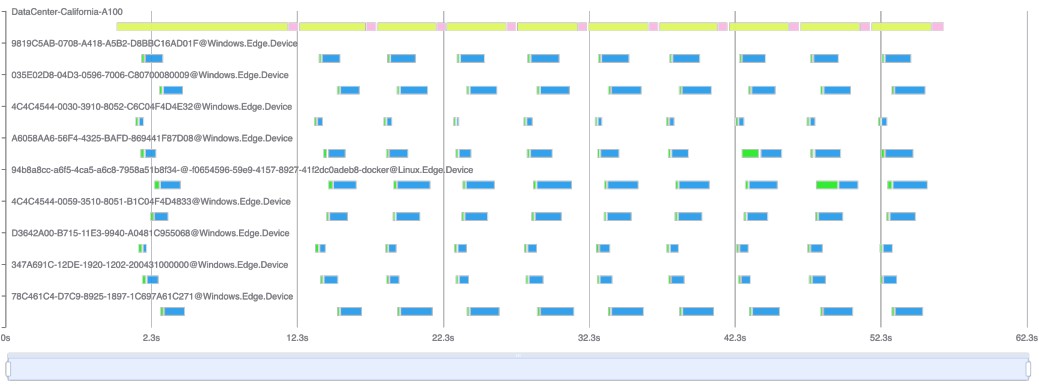

Figure 11: Real-world application demonstration. Yellow: aggregation server waiting time; pink: aggregation time; green: client training time; blue: client communication time.

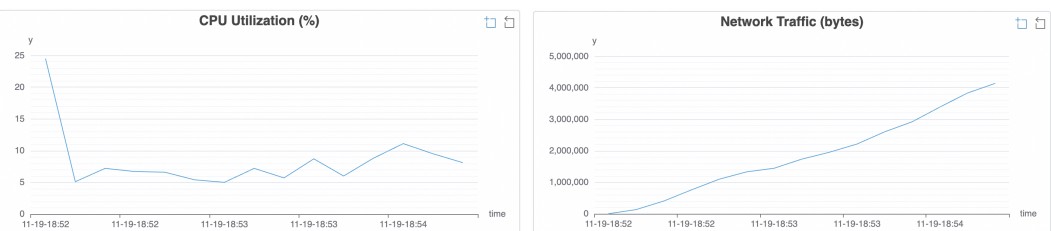

Figure 12: CPU utilization.        Figure 13: Network traffic.

## A.7 CODE IMPLEMENTATION

Implementations for anomaly detection can be found at `https://gitfront.io/r/user-5174596/LghVL3hZgZ34/anomaly-detection-code/`.

Implementations for zero-knowledge-proof verification can be found at `https://gitfront.io/r/user-5174596/eeZxuwaKtPnU/outlier-detection-zkp/`.

