# OpenReview forum: "Kick Bad Guys Out! Conditionally Activated Anomaly Detection in Federated Learning with Zero-Knowledge Proof Verification"
_ICLR.cc/2025/Conference — ICLR 2025 Conference Withdrawn Submission_

### Official Review · Reviewer_S7bc · 2024-10-31

**Soundness:** 2
**Presentation:** 3
**Contribution:** 2
**Rating:** 3
**Confidence:** 4

**Summary:**

The paper introduces an anomaly detection mechanism for federated learning, which activates the defense only when attacks are detected. The paper involves zero-knowledge proofs to ensure the integrity of the proposed defense mechanism.

**Strengths:**

1. The idea of using ZKP for FL defense is interesting.

2. The paper is well-organized and easy to follow.

**Weaknesses:**

1. The assumption that each local model and its reference model to be high seems to be only true under i.i.d. settings. Under non-i.i.d. settings

2. The paper does not provide any theorems to guarantee the defensive performance.

3. There is no table in the paper to clearly compare the results of different baselines and the proposed method under both i.i.d. and non-i.i.d. settings.

4. The paper does not use some advanced baselines such as FLTrust and FLDetector.

**Questions:**

Please refer to the weakness part.

---

### Official Review · Reviewer_2awL · 2024-10-31

**Soundness:** 2
**Presentation:** 3
**Contribution:** 2
**Rating:** 3
**Confidence:** 4

**Summary:**

The authors present a novel anomaly detection approach specifically designed for real-world FL systems. Their approach utilizes an early cross-round check that activates subsequent anomaly detection exclusively in the presence of attacks. When attacks occur, this approach efficiently removes malicious client models, ensuring that the local models submitted by benign clients remain unaffected. By leveraging ZKPs, the approach enables clients to verify the integrity of the removal performed by the server.

**Strengths:**

1.	The paper is well written and generally easy to follow.
2.	The authors conduct evaluations in a real-world setting.

**Weaknesses:**

1.	The proposed method relies on the assumption that malicious local models will deviate significantly from reference models (the previous global model and the client’s local model). However, this assumption only holds against weaker attacks. For more advanced attacks (such as LIE, MinMax, MinSum, and 3DFed), the differences between malicious and benign models are minimal, making detection extremely challenging. Unfortunately, the authors neither discuss these attacks in the Related Works section nor compare their method against them in the experiments.
2.	The authors evaluate only four weak Byzantine and backdoor attacks, all of which are relatively outdated, published before 2020, which is insufficient to demonstrate the effectiveness of the method. Furthermore, label flipping is not a backdoor attack. On the defense side, only five weak and outdated baselines were considered. I recommend that the authors include a broader set of stronger attacks and defenses.
3.	The default scenario of only considering 10 clients is unrealistic, as real-world FL settings generally involve a much larger number of participants, with current defenses typically tested on 100 clients or more. Although the authors do explore different client counts in Figure 4(d), most experiments still rely on the 10-client setup, which limits confidence in the method’s overall effectiveness.
4.	Several important evaluations are missing. For example, there is no analysis of whether the accuracy of the global model is affected in non-adversarial settings or whether the defense remains effective under attacker-dominated conditions. Additionally, the impact of varying levels of data heterogeneity has not been evaluated. Lastly, there is no ablation study or experimental analysis of robustness against adaptive attacks.
5.	Why are Cross-Round Detection and Cross-Client Detection effective? What is the convergence behavior of the proposed method? The authors should provide theoretical analysis for these aspects.

**Questions:**

Please refer to the Weaknesses.

---

### Official Review · Reviewer_k45M · 2024-11-01

**Soundness:** 3
**Presentation:** 3
**Contribution:** 3
**Rating:** 6
**Confidence:** 4

**Summary:**

Current defense methods are impractical for real FL systems because they rely on unrealistic prior knowledge or result in accuracy degradation even in the absence of attacks. Additionally, they lack verification protocols for execution, leaving participants uncertain about the correct implementation of mechanisms. To address these challenges, the authors proposed a new anomaly detection strategy that activates defenses only when potential attacks are detected, removing malicious models without affecting benign ones. Furthermore, they incorporated zero-knowledge proofs to ensure the integrity of the proposed defense mechanism.

**Strengths:**

The proposed method efficiently removes malicious client models when an attack occurs, without affecting the local models of benign clients. Additionally, it utilizes zero-knowledge proofs (ZKP) to enable clients to verify the integrity of the removal process conducted by the server.

**Weaknesses:**

The limitations of this study are not clearly defined. This study only addresses horizontal FL, with no mention of vertical FL.

**Questions:**

Does this study only address horizontal FL? Are there any limitations to applying it to vertical FL?

---

### Official Review · Reviewer_RAvJ · 2024-11-03

**Soundness:** 2
**Presentation:** 3
**Contribution:** 2
**Rating:** 3
**Confidence:** 4

**Summary:**

The paper presents a defense against poisoning attacks in federated learning settings capable of detecting and removing the effect of malicious participants. The proposed method relies on outlier detection, removing model updates with abnormal deviation with respect to reference models from previous training rounds. Two mechanisms are used for this purpose. First, a cross-round detection mechanism, where similarity scores with respect to previous reference models are computed and compared to a threshold. In a second step, a cross-client detection method is applied by removing those participants whose parameters differs significantly from the parameters of other clients for a given training round. To enhance the trust on the server side, the authors also propose to use a zero-knowledge proof technique to verify that the computations on the server can be trusted. The experimental evaluation on different benchmarks show that the method is capable of mitigating some poisoning and backdoor attacks in federated learning and to detect malicious participants successfully.

**Strengths:**

+ The proposed method aims not only to produce a model robust to poisoning attacks in federated learning (FL), but also to provide a series of characteristics that are not present in some robust methods in the research literature, such as detection of malicious participants, or trying to minimize the loss in accuracy of the resulting model when no attackers are present.

+ The paper includes a mechanism for checking the trustworthiness of the aggregator by using zero-knowledge proofs. This allows to check the integrity and honest execution of the algorithm in the aggregator and can have applicability in some scenarios where the aggregator may not be trusted.

**Weaknesses:**

+ The robustness of the proposed defense is only tested against attacks that typically do not consider detection mechanisms. It would be necessary to test the defense against more advanced attacks, including, for example, stealthy model poisoning attacks e.g., Bhagoji et al. “Analzying federated learning through and adversarial lens”, the poisoning attacks in Fung et al, Fang et al. “Local Model Poisoning Attacks to Byzantine-Robust Federated Learning”, or distributed backdoor attacks, to cite some. It would be also convenient to test the algorithm against adaptive attacks, but even if these are not considered, I believe the paper lacks an evaluation on more advanced attack scenarios.

+ Related to the previous point, the cross-client detection algorithm in Section 3.2, relies on the assumption that the malicious model updates are outliers. However, smart adversaries can try to create more subtle attacks in order to subvert the training of the FL algorithm without being detected (as some of the attack strategies mentioned before). The cosine similarity can be an interesting approach to defend against some attacks, like the ones used in the experiments, but present some limitations against more challenging attacks.

+ There are other papers in the related literature that aim to detect malicious clients. See for example, Li et al. “Learning to Detect Malicious Clients for Robust Federated Learning”, Muñoz-González et al. “Byzantine-Robust Federated Learning through Adaptive Model Averaging”, Zhang et al. “FLDetector: Defending Federated Learning Against Model Poisoning Attacks via Detecting Malicious Clients.” However, the authors did not compare to any existing method for detecting malicious participants in FL.

+ The strategy discussed in Section 3.3 for reducing the computational complexity, by choosing a layer for the reference models, opens the door to different types of attacks against the target model. Thus, attackers can modify freely parameters from other layers bypassing detection. It is unclear how the defense can overcome this type of situations.

+ The scalability of the proposed approach is not well discussed in the main paper. For example, storing the reference models from all participants in cross-device settings with hundreds of thousands of participants can be computationally very expensive.

+ Some of the results in the experiments see for example Figures 4 c) and d) and Figure 5, do not show the performance after model’s convergence. It would be necessary to run the experiments for more training rounds to have a more meaningful view of the model’s performance after training is completed.

**Questions:**

+ I do not think that the applicability of the 3-sigma rule to this case is not well justified, especially as the data distributions from participants is typically non-IID. Thus, there is no reason to support that the distribution of parameters sent by the participants follow a normal distribution, especially in some scenarios like the ones the authors used for testing (with 10 clients). In this sense, can the authors justify how the results in (Chang et al., 2006; of Public Health, 2001), cited by the authors to justify this point, apply in the case of participants having different types of non-IID distributions?

+ As for the method described in Section 3.3., how the defense can overcome the type of situations described earlier in my comments?

+ In algorithm 1, what happens when a client has not been selected by the server for a few number of training rounds? In this case, we can expect the model update to be quite different compared to the previous version of the local model stored by the server, and thus, trigger an anomaly. In this sense, for how long a reference model is valid?

+ In Section 5, the authors say: “By default, we use 10 clients for FL training, corresponding to real-world FL applications where the number of clients is typically less than 10”. Can the authors support this claim? Cross-device applications of FL use a much larger number of participants. See for example the case of Google Gboard.

+ How is the sensitivity computed for experiment 1?

+ In experiment 2, the cross-success rate provides some useful insights but fails to determine whether the anomaly detection proposed is affecting the performance negatively. Could the authors report the performance of the model as a function of the threshold?

---

### Note · Authors · 2024-11-13

I have read and agree with the venue's withdrawal policy on behalf of myself and my co-authors.